# Sedation of Wild Pyrenean Capercaillie (*Tetrao urogallus aquitanicus*) Using Intramuscular Midazolam

**DOI:** 10.3390/ani12141773

**Published:** 2022-07-11

**Authors:** Olga Nicolás Francisco, Ivan Afonso Jordana, Diego Garcia Ferré, Job Roig Simón, Ana Carolina Ewbank, Antoni Margalida, Irene Sacristán, Kévin Foulché, Emmanuel Ménoni, Carlos Sacristán

**Affiliations:** 1Forestal Catalana, Ministry of Climate Action, Food and Rural Agenda (Government of Catalonia), 25595 Lleida, Spain; 2Natural Environment Department, Conselh Generau d’Aran, 25530 Vielha, Spain; i.afonso@aran.org; 3Flora and Fauna Service, Department of Climatic Action, Food and Rural Agenda (Government of Catalonia), 08036 Barcelona, Spain; adgarfe@gencat.cat; 4Casa Bolunya S/N Seurí, 25567 Lleida, Spain; jobroig@yahoo.es; 5Laboratory of Wildlife Comparative Pathology, School of Veterinary Medicine and Animal Science, University of São Paulo, Sao Paulo 05508-010, Brazil; acarolewbank@gmail.com; 6Instituto Pirenaico de Ecología-Consejo Superior de Investigaciones Científicas, 22700 Jaca, Spain; a.margalida@csic.es; 7Animal Health Research Center, National Institute of Agricultural and Food Research and Technology—The Spanish National Research Council, 28130 Valdeolmos, Spain; isacristan.vet@gmail.com (I.S.); carlosvet.sac@gmail.com (C.S.); 8French Biodiversity Agency, Dead End de la Chapelle, 31800 Villeneuve de Rivière, France; kevin.foulche@ofb.gouv.fr (K.F.); emmanuel.menoni@wanadoo.fr (E.M.)

**Keywords:** capture myopathy, conservation, flumazenil, Galliformes, GPS tracking, movement ecology

## Abstract

**Simple Summary:**

Wildlife management often requires animal capture and handling, which may represent a challenge to animal survival after release. Capture myopathy has been extensively described as a consequence of wildlife capture and handling; however, despite its relevance, it may often be underestimated (i.e., carcass predation, absence of post-mortem examination). Although sedation can reduce management-related stress, little is known about sedation in wild Galliformes. Herein, we describe a sedation protocol using intramuscular midazolam in 23 wild Pyrenean Capercaillies (*Tetrao urogallus aquitanicus*) during field procedures (i.e., capture, handling, and tagging) in the Catalan Pyrenees, Spain. Our findings show that this sedation protocol is a valuable tool for the management of the endangered wild Pyrenean Capercaillies by promoting safer and easier capture and handling.

**Abstract:**

Global Positioning System (GPS) tracking offers key information in the study of movement ecology of threatened species. Nevertheless, the placement of GPS devices requires animal capture and handling, which may represent a challenge to the individual’s survival after release, mainly due to capture myopathy. The Pyrenean Capercaillie (*Tetrao urogallus aquitanicus*) is a threatened galliform especially sensitive to handling, extremely elusive, and challenging to capture. Our goal was to adapt a sedation protocol for Pyrenean Capercaillies undergoing GPS tagging, in order to increase their welfare and safety during the procedure. From 2018 to 2021, 23 wild Pyrenean Capercaillies were captured and sedated for GPS tagging as part of a European conservation project of emblematic Pyrenean avian species. The birds received intramuscular (IM) sedation with midazolam (ranging from 1.9 mg/kg to 8.08 mg/kg) and were handled for 20 to 40 min. Sedation was reversed with flumazenil (0.1 mg/mL IM). The sedated capercaillies were less responsive to stimuli (i.e., closed eyes and recumbency), showing discrete to no response to handling (i.e., placement of the GPS device, physical examination, cloacal temperature measurement, or reflex tests). Such response was compared in birds with sedation doses above and below the average dose (5.17 mg/kg). Only one clinical sign showed statistically significant differences between the two groups (“open-mouth breathing” sign, *p* = 0.02). A mortality rate of 4.35% was registered (one individual died during handling). Sedation facilitated the handling of the birds and faster interventions in the field, without increasing mortality when compared to handling without sedation. Therefore, sedation was shown to be a useful tool to reduce stress related to capture and handling of the threatened Pyrenean Capercaillie.

## 1. Introduction

The Western Capercaillie (*Tetrao urogallus*, order Galliformes) is the largest European grouse species [1], currently listed as “Least concern” (IUCN, year) despite its decreasing population trend [1]. In contrast, the subspecies Pyrenean Capercaillie (*T. u. aquitanicus*), inhabitant of the Pyrenees, is currently considered “Vulnerable” in Spain, a status that will soon be changed to “Endangered” due to a significant population decline trend [2,3]. The main threats to the Pyrenean Capercaillie include climate change, habitat fragmentation, flight collisions (e.g., fences, ski lifts, and electric cables), competition with ungulates, and anthropogenic activities [4,5,6,7].

Movement ecology is a valuable tool in the conservation of threatened species, useful for understanding movement patterns, demographic parameters, and land use [8,9], based on animal capture and handling for the placement of Global Positioning System (GPS) harnesses and collars. Captures of Pyrenean Capercaillies from 2007 to 2021 (without sedation nor vitamin E and selenium administration) were related to high mortality rates during handling and within the following four weeks 10 of 59 (17%) birds captured in Catalonia and 6 of 29 (20.69%) birds captured in France; Government of Catalonia and Office Français de la Biodiversité (unpublished data). Mortality related to capture events has been reported in other wild avian species, i.e., 30.4% mortality of Red-legged Partridge (*Alectoris rufa*) was observed within 72 h following net capture due to capture myopathy and self-injuries [10]; 36% mortality within 12 days following capture and tagging of 33 Pileated Woodpeckers (*Dryocopus pileatus*) [11], and 100% mortality within 16 days of capture and tagging of 4 Little Bustards (*Tetrax tetrax*) [12]. Additionally, studies in other wild bird species (i.e., Wild Turkey *Meleagridis gallopavo*, Sandhill Crane *Grus canadensis*, Short-tailed Albatross *Phoebastoria albatrus*, Rock Ptarmigan *Lagopus muta pyrenaica*) described mortality peaks within a few weeks following stressful situations, such as short handling periods, translocations, or reintroductions [13,14,15,16]. In such cases, the related cause of death was often identified as natural mortality, underestimating the influence of complications linked to capturing procedures, such as capture myopathy (CM) [17]. Capture myopathy is a well-known complication associated with the stress caused by capturing and handling procedures, capable of hampering the survival rate of wildlife after their release [10,14,17,18,19,20,21], and therefore, an important differential diagnosis. CM-affected animals may be slower and less alert, or even suffer behavior impairment, pain or inability to move, becoming more susceptible to predation [17]. The main CM consequence is immediate or delayed mortality (within several weeks following capture) [10,17].

Sedation with benzodiazepines, alpha-agonists, opioids, or combinations of some of these drugs has been widely used in avian clinical practice [22,23,24,25]. Additionally, it has demonstrably reduced capture- and handling-related stress in birds [26,27,28,29,30,31,32,33]. Midazolam, a short-acting and water-soluble benzodiazepine with anxiolytic, muscle relaxant, sedative, and anticonvulsant effects [24], has proven particularly useful [31,32]. Additionally, protocols relying on midazolam are particularly useful in wildlife field procedures due to the possibility of reversal (i.e., flumazenil), promoting safer and reversible procedures. Nevertheless, there are no previous sedation studies in threatened wild Galliformes. In order to address this gap of knowledge and further contribute to conservation studies on Pyrenean Capercaillie, we adapted a sedation protocol previously established in Grey Partridge (*Perdix perdix*) [30] to improve welfare and safety in wild Pyrenean Capercaillie undergoing GPS tagging.

## 2. Materials and Methods

### 2.1. Captured and Sedated Animals

A total of 23 wild Pyrenean Capercaillies (19 adults and 1 juvenile male and 3 adult females) were captured in the Catalan Pyrenees (in the counties of Pallars Sobirà, Vall d’Aran, Alt Urgell, Ripollès, Cerdanya and Berguedà; Catalonia; Spain), between 2018 and 2021 (Table 1).

### 2.2. Capture and Handling

Most captures (*n* = 21) were conducted in May, the month in which Pyrenean Capercaillie males perform courtship and mounting behavior in leks—natural clearings in black or red pines forests. During the study, from 2018 to 2021, the mean temperatures in most of the capercaillie distribution areas in May were on average 8 to 10 °C and, in September, 10 to 12 °C.

The majority of the studied birds were captured after becoming entangled in nets of approximately 1–1.5 m high (Figure 1), placed in the leks [34]. All captures occurred at sunset or sunrise, when males and females are more active. The remaining captures (*n* = 2) were performed in September, using an approximately 50 cm-high mesh placed in order to form a walk-through passage towards a trap cage (“lily trap”) that was programmed to automatically send a photograph when activated.

### 2.3. Sedation and Handling

Subsequently, and while the capercaillie was still entangled in the net or in the cage, a veterinarian quickly palpated the keel to ensure a good nutritional condition and administered midazolam (Midazolam B Braun 5 mg/mL, B. Braun Melsungen AG, Melsungen, Germany) with a 23-gauge needle into the major pectoral muscles. The light stimuli were immediately halted by a tube-shaped adapted fabric hood that allowed the bird to breath normally and prevented corneal lesions. The time interval between the capercaillie capture (first contact with the net or entrance in the trap) and the sedative administration (henceforth, “capture time”) was kept to the strict minimum time. Thus, the syringe containing the sedative and all other necessary materials were prepared prior to the capture event and maintained at hand. The birds were manually restrained for 2 to 3 min to promote proper sedation and to avoid unconscious flapping movements during induction (“induction time”, time period between midazolam administration and the first signs of decreased awareness/consciousness and relaxation). Once the first signs of sedation were observed (i.e., muscle relaxation and head dropping), the bird was untangled from the net or released from the cage. The midazolam injection time was considered as the onset of the “handling time” (Ht). During the following 20 to 30 min, the GPS harness was placed, and biometrical measures were obtained. For this sedation protocol, the capture team set up a maximum Ht of 40 min. Additionally, physical examination was performed by a veterinarian to evaluate the bird’s overall condition and health status (Appendix A): body score (i.e., from 1—emaciated, to 5—obese) [35,36], weight (with a digital hand dynamometer scale), heart and lung auscultation and frequency, cloacal temperature, and inspection of the ocular and oral mucosa, cloacal area, feet, and feathers. Selected reflexes and responses were also tested during Ht in order to assess sedation depth (i.e., palpebral reflex, wing and leg withdrawal responses, and deep pain or nociception assessment through toe pinching, presence of leg withdrawal reflex, and cognitive body movement) [37,38]. Other clinical signs, movements, or reactions (i.e., head position, rigidity, wing flapping, and tremors) were also registered. Early on into Ht, vitamin E/selenium combination (Esvex^®^, S.P. Veterinaria, S.A., Tarragona, Spain) was administered IM, due to its reported ability to prevent capture myopathy in other wild avian species [16,17,39,40].

### 2.4. Reversal and Release

Following GPS placement, physical examination, biometry, and blood sampling (data not shown), the sedated bird was translocated to the release site, i.e., a flat spot in the forest, a few meters away from the capture site, to allow the bird to recover quietly and under minimum noise stimuli. Once at the release site, flumazenil (Flumazenil Altan 0.1 mg/mL, Altan Pharmaceuticals, S.A., Madrid, Spain) was administered to reverse sedation (end of Ht). In 2018, flumazenil was administered IM because it was considered an easier route to use in field studies and due to previous reports [41,42,43]. Nevertheless, since 2019, flumazenil has been administered as an intravenous slow infusion into the brachial vein to promote faster reversal. When possible, the bird was partially covered with understory vegetation at the chosen release site or with an additional dark blanket or sheet to promote a less stressful reversal (i.e., reducing the light stimuli). This method ensures a smoother reversal period, preventing the flee response of flying or walking when the birds are still unbalanced.

### 2.5. GPS Devices and Post-Procedure Monitoring

The GPS devices used to tag the birds were placed as backpacks [44] (Figure 2) or pelvic harnesses [44,45], weighting between 10 and 45 gr. The used technologies were GPS–GSM (Global Positioning System–Global System for Mobile communication) and GPS–VHF of selected commercial brands (Ornitela–Ornitela, UAB, Lithuania, Milsar -Milsar Technologies S.R.L., Romania and Lotek, UK).

All the individuals included in this study were monitored for weeks to years following their capture. Additionally, the GPS devices were programmed to obtain detailed position and posture information for 24–48 h after release. When available, the carcasses of the birds that died during or after the procedures went through gross and histopathologic examination [45].

### 2.6. Statistics

We used the Kruskal–Wallis test to compare the observed mortality in sedated animals from this study, during handling and within four weeks post procedure, with mortality in non-sedated animals during the same time frame, as recorded in previous captures of Pyrenean capercaillies in Spain and France (1,2). The Fisher’s exact test was used to compare the mean number of physical events (palpebral reflexes, wing and leg withdrawal, upright head position, deep pain response, rigidity, body arching, open mouth breathing, and sporadic wing flapping) in animals that received a sedation dose above and below the average dose used in this study (5.17 mg/kg). The significance value was *p* = 0.05. The statistical software and package used for the statistical analysis was GraphPad Prism version 6.04 for Windows, GraphPad Software, La Jolla, CA, USA.

## 3. Results

### 3.1. Sedated Pyrenean Capercaillies: Captures, Doses, and Cloacal Temperatures

In 2018, six birds were sedated with midazolam (Table 1). The first administration, in a male Pyrenean Capercaillie (8.09 mg/kg), induced deep sedation (absence of reflexes responses and deep pain reaction) and absence of head stance and wing flapping.

In the following captures and subsequent years, the midazolam dose was gradually decreased, to administer the minimum sedative dose necessary to achieve light sedation and promote a fast and uneventful recovery. In 2019 and 2020 (*n* = 10), the dose was lowered to 4.5 mg/kg. Finally, in 2021 (*n* = 7), the birds received an even lower midazolam dose (down to 2.8 mg/kg). The average midazolam dose administered to the 23 Pyrenean Capercaillies was 5.17 mg/kg. Midazolam doses according to the year and place of capture, weight, and sex are shown in Table 2. Additionally, during the first 10 handling min, vitamin E/selenium was administered IM (30 mg/kg vitamin E and 0.1 mg/kg selenium). Cloacal temperatures were registered in 11 birds between 3 and 30 min after midazolam administration. The temperatures ranged from 39.9 °C to 41.9 °C (mean temperature: 40.9 °C) (Table 2).

### 3.2. Capture Time, Induction Time, Ht

The capture time ranged between less than 60 s and up to 240 s. The induction time varied between 60 s and 300 s. Ht lasted between 20 and 30 min, with the exception of two cases, for which it went beyond the theoretical maximum of 40 min (43 and 56 min, respectively).

### 3.3. Sedation Parameters and Physical Examination

Following induction, the birds were recumbent and relaxed and presented little to no wing flapping (Figure 2). All parameters evaluated on physical examination were within the normal limits, and there were no changes on heart and lung auscultation. The birds also presented good body and feather condition. On physical examination, there were no relevant lesions, and their nutritional condition was considered excellent. Only one adult male presented skin lesions compatible with breeding disputes (small hematomas and skin crusts). The weight was registered for 18 birds (Table 1).

Sedation response and depth were evaluated based on the results of several tests: palpebral reflex test (positive in 2 out of the 19 birds tested), wing and leg withdrawal reflex (present in 3 out of the 19 birds tested), and deep pain response (observed 1 out of the 19 birds tested) (Table 2). Additionally, 16 capercaillies presented an upright head position (23 birds tested), indicative of a light sedation plane and apparently most commonly observed with lower sedative doses. Rigidity, body arching, open-mouth breathing, and sporadic wing flapping were sporadically detected during Ht in nine cases (Table 2).

In order to compare birds’ responses and test results according to the sedation dose, the sedated birds were divided into two groups: those that received a dose below (5.17 mg/kg) and those that received a dose above the average dose (Figure 3). The only sign that showed statistically significant differences between these two groups was the “open-mouth breathing” sign (Fisher’s exact test, *p* = 0.02).

### 3.4. Sedation Reversal

Flumazenil doses ranged from 0.06 mg/kg to 0.12 mg/kg IM, directly proportional to the midazolam dose each individual received. Despite reversal, 15 Pyrenean Capercaillies remained at the release site after flumazenil administration. Two of them flew away the day after capture, when a passer-by approached the release site. Furthermore, three Pyrenean Capercaillies in 2018 and another one in 2021 tried to fly away before they were fully recovered and coordinated (i.e., rolling over, wing flapping).

### 3.5. Losses

Starting at capture and within the four weeks in which the birds were tracked, only one death was registered (1/23; 4.35%). The dead individual was an adult male captured in the lek in 2020, that died within the first minutes of handling (Capture time = 60 s, Ht = 300 s). The bird received a dose of 4.73 mg/kg IM midazolam. It presented good body condition (score 4), stress bars on the tail feathers, and marked lice infestation. Post-mortem examination revealed large and recent subcutaneous and muscle hematomas in the medial aspect of both knees and inguinal areas. Histopathologically, the main observed lesions were systemic congestion, pulmonary edema, multifocal interstitial hemorrhage of the cardiac apex, multifocal radbomyonecrosis, multifocal perivascular splenic hemorrhage, adrenal hypertrophy, and diffuse vacuolization of superficial, and deep interrenal cells. Additionally, microgranulomatous lesions were observed in the lungs, likely due to nematode migration.

Mortality detected in the sedated birds included in this manuscript (4.35%, 1/23) was compared to mortality observed in non-sedated birds in Catalonia (17.0%, 10/59) and France (20.7%, 6/29). We detect a tendency of mortality reduction in sedated Pyrenean Capercaillies after capture and handling of this species, although not statistically significant differences were obtained (Kruskal–Wallis test, *p* = 0.07) (Figure 4).

### 3.6. GPS Devices and Post-Capture Follow-Up

The GPS positions showed that the sedated capercaillies stayed at the release site during a mean time of 18 h and 8 min (*n* = 13). Their first movement was, on average, of 260 m from the release site (*n* = 15). The sedated Pyrenean Capercaillies were tracked by a mean period of 398 days (ranging from 89 to 1226 days), until May 2022, and eight individuals are still carrying functioning GPS devices.

Additionally, no predation occurred after Pyrenean Capercaillie capture and sedation, as confirmed by the GPS positioning of the tagged birds.

## 4. Discussion

We found that sedation of Pyrenean Capercaillies with IM midazolam was effective, reducing animal consciousness and facilitating handling, thus allowing faster and more efficient interventions in the field. Midazolam was already described to improve handling tolerance in other wild avian species [46]. Using sedation in wild Pyrenean Capercaillie implies working with relaxed animals showing no resistance to handling. As a result, sedated Pyrenean Capercaillies may be less prone to suffering stress-associated lesions, as described in midazolam-sedated Bar-tailed Godwit (*Limosa lapponica*) [46].

Sedation depth varied according to the protocols used in different years. A higher dose of midazolam (8.09 mg/kg) promoted deeper sedation (i.e., more relaxed birds, head dropping, no wing flapping) than a lower dose (i.e., 2.88 g/kg), as observed in previous studies in wild birds [29,32]. We recommend a light sedation for the type (not painful) and length (20 to 30 min) of handling (GPS tagging) performed herein in Pyrenean Capercaillie. Such light sedation can be achieved with the administration of the lower IM midazolam volume: 2 to 3 mL in males (dose of 2.7 mg/kg to 4.05 mg/kg for an adult male of 3.7 kg in weight) and 1.5 mL in females (average dose of 4.16 mg/kg for an adult female of 1.8 kg). As reported above, it is crucial to keep the capture time to a minimum; thus, the syringe containing midazolam must be prepared just before capture (we recommend preparing at least one syringe for males and another one for females). Furthermore, reducing the handling time has also been outlined as a goal when working with wild avian species [11,15].

The hyperacute death of an adult male occurred during handling. Certain histopathological findings (e.g., multifocal interstitial hemorrhage in the heart apex, pulmonary edema, systemic congestion, multifocal perivascular spleen hemorrhage, multifocal radbomyonecrosis in a skeletal muscle) suggest that the captured capercaillie likely suffered an acute shock, while the interrenal cell lesions could be attributed to chronic stress. Some of these lesions (i.e., necrosis in the skeletal muscle) have been previously associated with CM [11,12,15], but one cannot discard other concomitant causes, i.e., chronic disease or lek fights between males (muscle hematomas). Additionally, some unexpected and rare reactions (i.e., rigidity and arching of the head and body, open-mouth breathing) were observed in three sedated males. Similar responses (i.e., agitation, nystagmus, reaction to restraint) were described in two sedated Mangrove Parrots (*Amazonica amazonica*) after administration of intranasal and intramuscular midazolam [47]. These authors suggest that these responses resembled those defined as paradoxical reactions to benzodiazepines [47], well-known in dogs and cats [48].

Midazolam pharmacokinetics reports in poultry describe a plasma half-life ranging from 0.42 to 9.71 h after IM administration [49]. When sedation was not reversed, the length of midazolam sedation with a 1.5 mg/kg dose in other wild avian species varied from 1 to 3 h in wild Bar-tailed Godwits [46] and from 2 to 3 h in Ring-necked Parakeets (*Psittacula krameri*) [24]. Faster sedation reversals are achieved when flumazenil is administered. For instance, in quails sedated with 6 m/kg IM midazolam, the authors observed complete reversal following administration of 0.1 mg/kg IM flumazenil in less than 2 min, without any resedation signs [32]. The recovery of Pyrenean Capercaillies after the administration of flumazenil was not immediate. The administered flumazenil doses (ranging from 0.06 to 0.12 mg/kg) were very similar to those recommended for pet birds, i.e., from 0.01 to 0.1 mg/kg [43]. GPS positioning revealed that 13 individuals remained at the release site during an average time of approximately 18 h. Two birds remained at the release site until approached by a passer-by, 24 h post-capture. Such information was recorded after the GPS data analysis, once the birds had already moved away from the release site. Thus, one cannot affirm if this lack of movement occurred because the birds were not fully recovered yet or because they were unable to move normally after sedation. In spite of that, no predation was detected after capture and sedation.

There were only three females among the sedated Pyrenean Capercaillies between 2018 and 2021. The low proportion of females captured from 2017 to 2021 may be due to the fact that most of the captures were conducted in the lek, where females spend only a few hours during the heat season peak.

Midazolam has shown to cause relaxation and high tolerance in sedated birds to handling, proving to be a valuable tool to warrant Pyrenean Capercaillie welfare during capture and handling. The cloacal temperatures measured in sedated capercaillies were within the normal temperature range for avian species [50]. It has been previously reported that preventing hyperthermia may protect animals from developing capture myopathy [20,21]. We believe that this sedation protocol provides a useful tool to prevent stress and therefore reduces the risk of CM associated with capture and handling. There were no statistical significant differences regarding mortality between sedated and non-sedated Pyrenean Capercaillies, likely due to the relatively small sample size of sedated birds. However, we observed a tendency of mortality reduction in sedated birds when compared to non-sedated birds. Regardless, previous studies in other wildlife species (artiodactyls) have recommended interrupting captures when the mortality rate is greater or equal to 2% during captures [51].

The lack of statistical analyses on complete physical evaluations and neurological tests between sedated and non-sedated individuals is due to the absence of comparable/recorded data in non-sedated capercaillies from previous captures, which is a study limitation.

Future captures should work towards increasing the number of studied animals in order to provide further data on the parameters evaluated herein, especially regarding female sedation. Additionally, an effort should be made to improve, shorten, and increase the control over the recovery time and reversal of the birds, further calibrating the ideal doses and promoting even smoother and faster recoveries.

## 5. Conclusions

Sedation of wild threatened Pyrenean Capercaillie with IM midazolam has shown to be an effective procedure to decrease the birds’ response to stimuli and increase their welfare and safety during handling. The improved welfare observed with our sedation protocol could be a useful tool for threatened galliform management and conservation. The sedated birds were more relaxed, less sensitive to the stressors implied in the tagging process, and less resistant to handling than non-sedated Pyrenean Capercaillies, which reduced the risk of accidental injuries. Additionally, field technicians were able to work more comfortably and faster. So far, a volume of 2 to 3 mL IM midazolam for males (average dose of 2.7 mg/kg to 4.05 mg/kg for an adult male of 3.7 kg in weight) and 1.5 mL for females (average dose of 4.16 mg/kg for an adult female of 1.8 kg) apparently provides the light sedation plane required for non-painful and short handling procedures, i.e., tagging Pyrenean Capercaillies with GPS devices. The adaptation of our sedation protocol could be useful for threatened galliform management and conservation.

## Figures and Tables

**Figure 1 animals-12-01773-f001:**
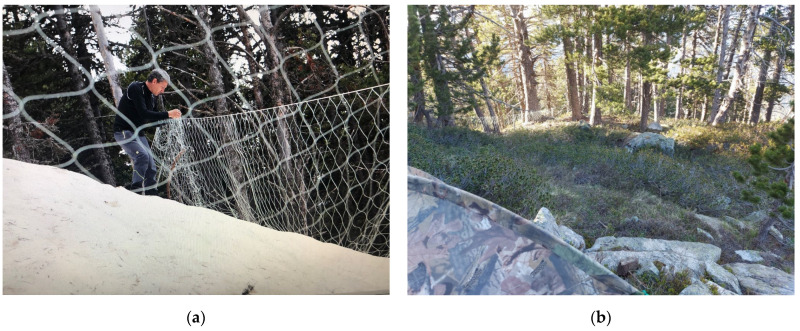
(**a**) Placement of a capture net in Val d’Aran. Image Source: Javier Montes (**b**) Hide close to the capture net in Val d’Aran.

**Figure 2 animals-12-01773-f002:**
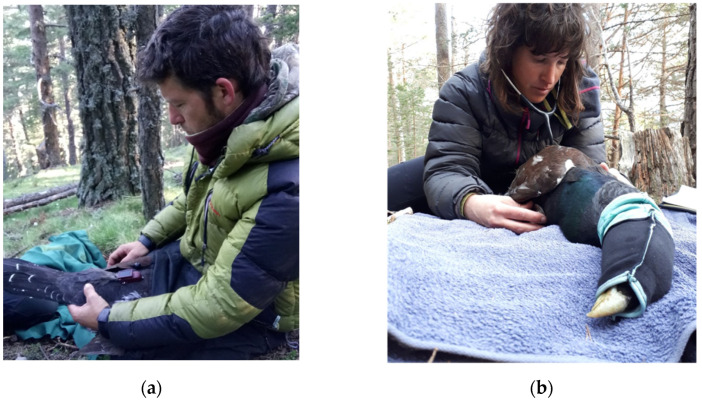
(**a**) Placement of the GPS device; (**b**) Heart auscultation of a sedated and hooded Pyrenean capercaillie *(Tetrao urogallus aquitanicus*) (note: the bird is deeply sedated and requires no physical restraining). Image Source: Jordi Camprodon.

**Figure 3 animals-12-01773-f003:**
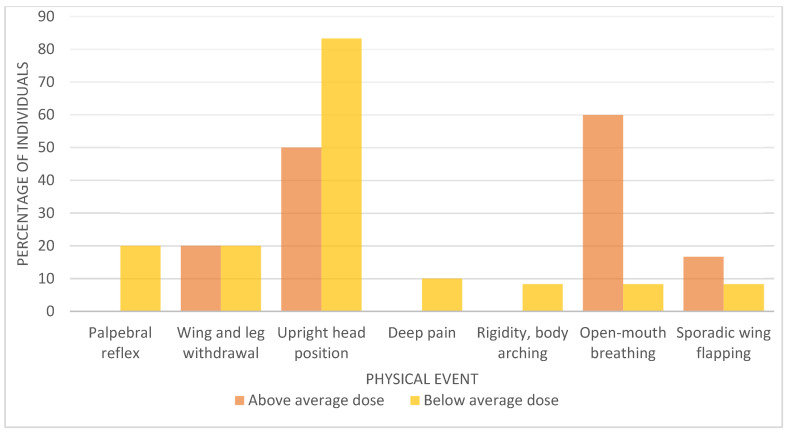
Comparison of the percentage of Pyrenean Capercaillies (*Tetrao urogallus aquitanicus*) in each group (group of animals administered a sedation dose above or below the average dose applied in the study (5.17 mg/kg)) that presented the described physical events.

**Figure 4 animals-12-01773-f004:**
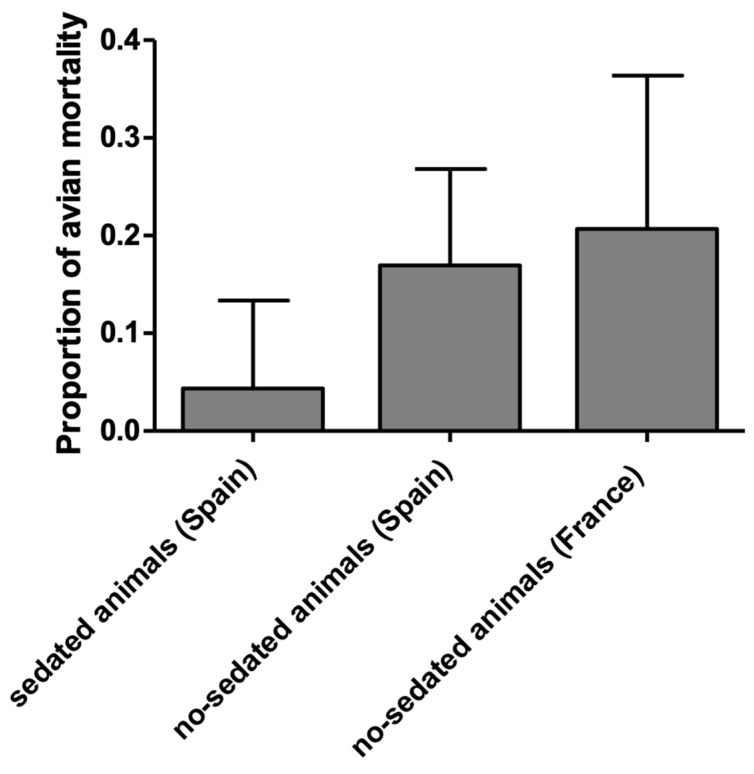
Proportion of mortality observed in sedated and non-sedated Pyrenean Capercaillies (*Tetrao urogallus aquitanicus*) from Spain and France. Kruskal–Wallis test *p* = 0.24.

**Table 1 animals-12-01773-t001:** Capture year, weight, sex, sedative volume (midazolam), sedative dose, reversal dose (flumazenil), reversal time, and cloacal temperature of the 23 sedated Pyrenean Capercaillies (*Tetrao urogallus aquitanicus*).

Case Number	Year	Weight	Sex	Sedative Volume (mL)	Sedative Dose (mg/kg)	Cloacal Temperature (°C)	Reversal Dose (mg/kg)	Reversal Time (min) ^1^
1	2018	3400	M	5.5	8.09	40.3	0.09	35
2	2018	3750	M	5.8	7.73	-	0.10	32
3	2018	3650	M	5	6.85	-	0.10	30
4	2018	1850	F	1.5	4.05	-	0.10	26
5	2018	3550	M	3.5	4.93	-	0.10	33
6	2018	N.R.*	M	5	-	-	-	NR
7	2019	3080	M	2.9	4.71	40	0.10	35
8	2019	2900	M	3.5	6.03	41.1	0.10	56
9	2019	3900	M	3.5	4.49	-	0.09	35
10	2019	3650	M	3.5	4.79	39.9	0.08	43
11	2019	3770	M	5	6.63	-	0.08	26
12	2020	3600	M	3.5	4.86	-	0.09	25
13	2020	3700	M	3.5	4.73	-	0.08	28
14	2020	3700	M	3.5	4.73	-	- ^2^	-
15	2020	1750	F	2.1	6	-	0.12	39
16	2020	2750	M	3	5.45	41.2	0.11	30
17	2021	3750	M	3	4	-	0.09	20
18	2021	3700	M	3	4.05	40.8	0.08	22
19	2021	N.R.	F	1.5	-	40.9	-	30
20	2021	N.R.	M	2	-	41.9	-	24
21	2021	3500	M	2	2.86	41.3	0.06	25
22	2021	N.R.	M	2	-	41.1	-	23
23	2021	2600	M	1.5	2.88	40.7	0.08	30

* N.R. = Not registered ^1^ Reversal time (min) = interval between sedative administration and reversal administration (in minutes). ^2^ Died during handling (not reversed).

**Table 2 animals-12-01773-t002:** Reflexes, wing and leg withdrawal responses, upright head position, deep pain response, and other observations (i.e., open-mouth breathing and sporadic wing flapping or tremors) registered on the sedated birds.

Case Number	Sex	Sedative Dose (mg/kg)	Palpebral Reflex	Wing and Leg Withdrawal Reflexes	Upright Head Position *	Deep Pain	Rigidity, Body Arching	Open-Mouth Breathing	Sporadic Wing Flapping or Tremors
1	M	8.09	no	no	no	no	no	yes	no
2	M	7.73	no	no	yes	no	no	yes	no
3	M	6.85	no	no	no	no	no	yes	no
4	F	4.05	yes	yes	yes	yes	no	no	no
5	M	4.93	no	no	no	no	yes	no	no
6	M	-	-	-	no	-	yes	no	no
7	M	4.71	yes	yes	yes	no	no	no	yes
8	M	6.03	no	no	no	no	no	no	yes
9	M	4.49	-	-	no	-	no	no	no
10	M	4.79	-	-	yes	-	no	yes	no
11	M	6.63	-	-	yes	-	-	-	no
12	M	4.86	no	no	yes	no	no	no	no
13	M	4.73	no	no	yes	no	no	no	no
14	M	-	no	no	no	no	no	no	no
15	F	6.00	no	yes	yes	no	no	no	no
16	M	5.45	no	no	yes	no	no	no	no
17	M	4.00	no	no	yes	no	no	no	no
18	M	4.05	no	no	yes	no	no	no	no
19	F	-	no	no	yes	no	no	no	no
20	M	-	no	no	yes	no	yes	yes	no
21	M	2.86	no	no	yes	no	no	no	no
22	M	-	no	no	yes	no	no	no	no
23	M	2.88	no	no	yes	no	no	no	no

* No upright head position = hanging head position.

## Data Availability

Data is contained within the article.

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
