# Peer review of "Sedation of Wild Pyrenean Capercaillie (Tetrao urogallus aquitanicus) Using Intramuscular Midazolam"

_animals, 2022, doi:10.3390/ani12141773_

Round 1

Reviewer 1 Report

The manuscript describes the experience of handling of wild Pyrenean capercaillie after capture. Midazolam was applied intramuscularly to reduce the stress of the capture. The doses were decreased and their effect was tested with aim of recommendation of the lowest dose which can cause acceptable sedation and fast recovery. The manuscript is clear and well organized. It can be of help for veterinarians who works with wild Galliformes.

The introduction presents clearly the problem of capture of wild birds, including Pyrenean capercaillie.

Material and methods are presented with enough information. It would be nice if mean temperature during May and September is given in the text (lines 104-111).

The authors should provide information in material and methods how was evaluated the body weight of the birds and how was calculated the administered dose of midazolam in field conditions.

Table 1 should be cited for first time in materials and methods.

Conclusion: It will be better to provide not only information for the volume of midazolam solution but also the relative dose. Flumazenil administration and its dose should be noticed.

Author Response

We would like to thank the Reviewer 1 for dedicating time to consider our manuscript entitled “Intramuscular midazolam for the sedation of wild Pyrenean Capercaillie (Tetrao urogallus)” (Animals Manuscript ID: animals-1747897) submitted for consideration in Animals. We appreciate very much all the comments that substantially contributed to improve our previous version. Herein we provide point-by-point answers, which have been fully considered and answered accordingly.

Kind regards,

Olga Nicolas (In behalf of all the authors)

Reviewer 2 Report

Reviewer comments on Manuscript number: animals- 1747897

The present manuscript shows the effect of intramuscular Midazolam for handling wild Pyrenean capercaille (Tetrao urogallus aquitanicus)

The manuscript is well written, and the information showed is interesting and relevant, as its main aim is to provide current information about the sedative effects of intramuscular midazolam for handling wild Pyrenean capercaillie.

Broad comments

The manuscript is well written, and the information showed is valuable and relevant, just few comments and suggestions before it could be published.

To this reviewer the title that better describes the study could be ”Evaluation of intramuscular Midazolam as a chemical restraint for wild Pyrenean capercaillie (Tetrao…….

Introduction.

It is important to add some information about other drugs that are feasible for handling wild birds. Why to choose midazolam among others?

Materials and methods

L132. Do you mean heart rate and respiratory rate? Please rephrase.

L135. Please specify the process of deep pain assessment.  

L144. What was the intended dose of flumazenil in mg/kg?

Table 1. Please add the flumazenil dose and the time (after midazolam administration) when it was administered in each case.

Results.

This section is particularly lacking of information about the time for onset of sedation and for complete recovery after administration of flumazenil. Considering the type of report this information is very important and has to be added.

Discussion

This section should consider the use of other drugs than midazolam for chemical restraint, the use of flumazenil antagonist and the time lapsed from sedation to full recovery.

Conclusions

This section should be shortened and specifically focused in what was found in this study. As no information was declared about the time lapsed from recovery after administration of midazolam; no conclusions can be made about the use of flumazenil to short the recovery time.

Thank you.

Author Response

We would like to thank the Reviewer 2 for dedicating time to consider our manuscript entitled “Intramuscular midazolam for the sedation of wild Pyrenean Capercaillie (Tetrao urogallus)” (Animals Manuscript ID: animals-1747897) submitted for consideration in Animals. We appreciate very much all the comments that substantially contributed to improve our previous version. Herein we provide point-by-point answers, which have been fully considered and answered accordingly.

Kind regards,

Olga Nicolas (In behalf of all the authors)

Reviewer 3 Report

It is overall a very interesting paper with impact on conservation. There are some minor spelling that needs to be improved. 

A more detailed summary of the table 2 would be good. Where certain clinical signs associated with higher/lower doses of midazolam? Provide mean and standard deviation for capture time, induction time and handling time for each of the different dosages. 

Flumazenil has a short half life and re-sedation has been observed after IV administration in birds. The authors should include this in their results and also discuss the dosage of flumazenil and birds remaining at capture site.

The reference need to be checked carefully different font and sizes throughout

See below for minor comments/spelling:

Minor changes:

Line 120- mini time provide mean and SD

Line 177- dosage of vitE/sel

Table 2: legend not se;f expalinary please add species and number of birds

Line 237 rhabdomyonecrosis

Line 288 add number of males in brackets

Reference

Ref 42 remove

Ref 47 no publisher, please add

Ref 38 – two references together

Years of publication highlighted in some bit not others

Ref 31- unpublished so should not be included

Ref 27 full stop missing

Ref 3 provide English translation 

Author Response

We would like to thank the Reviewer 3 for dedicating time to consider our manuscript entitled “Intramuscular midazolam for the sedation of wild Pyrenean Capercaillie (Tetrao urogallus)” (Animals Manuscript ID: animals-1747897) submitted for consideration in Animals. We appreciate very much all the comments and the correction, and have edited the manuscript accordingly.

Kind regards,

Olga Nicolas (In behalf of all the authors)

Reviewer 4 Report

This paper appears well written, with useful information regarding sedation doses, side effects and reversal information for wild galliformes to reduce capture myopathy associated mortality. The strengths of this paper include good sample size of birds that were sedated, with a range of doses used and the differences between doses discussed. Furthermore, the GPS tracking allowed follow-up of the population to assess post-sedation mortality. The one bird that passed away, had a post-mortem exam with histopathology to determine cause of death.

Weaknesses of this paper include lack of discussion regarding control birds, i.e. those that were studied in previous years, but were not sedated for capture. The authors mention unpublished data, can they expand on this? Furthermore, were these birds also treated with Vitamin E/Selenium injections, did this have an effect on their mortality rates?

Minor error with references list; between 38 and 39 there is an additional reference that is unnumbered (Mans, Sachez-Migallon et. al) 

Author Response

We would like to thank the Reviewer 4 for dedicating time to consider our manuscript entitled “Intramuscular midazolam for the sedation of wild Pyrenean Capercaillie (Tetrao urogallus)” (Animals Manuscript ID: animals-1747897) submitted for consideration in Animals. We appreciate very much all the comments that substantially contributed to improve our previous version. Herein we provide point-by-point answers, which have been fully considered and answered accordingly.

Kind regards,

Olga Nicolas (In behalf of all the authors)

Round 2

Reviewer 2 Report

Reviewer comments on Manuscript number: animals- 1747897 v2

The present manuscript shows the effect of intramuscular Midazolam for handling wild Pyrenean capercaille (Tetrao urogallus aquitanicus)

Broad comments

The manuscript has been improved, just few comments and suggestions before it could be published.

Abstract.

The aim of the study is missing and has to be added at the beginning of the abstract. The abstract should include the percentage and number of birds that died after the chemical restraint with intramuscular midazolam.

Introduction.

Information about other drugs that are feasible for handling wild birds is still missing and has to be added. Even when midazolam has apparently been useful; there are some other drugs that have been employed in the past but not necessarily safer. Two lines about those drugs are needed in order to introduce the reader to midazolam use.

The aim of the study must be clearly stated at the end of the introduction section.

Materials and methods

L143-144. Please specify and add the entire process of deep pain assessment (not briefly). 

L154. What was the intended dose of flumazenil in mg/kg? In this section it is important to add the dose in mg/kg. No matter the weight, the dose per kilogram of body weight must be added in this paragraph

Statistics

L 179-185. Please add the statistical software and package used for analysis.

Thank you.

Author Response

We thank reviewer 2 for all the comments and suggestions. We have answered to all of them in the rebuttal letter (attached as a Word file).

Reviewer 3 Report

Line 197 replace is with was 

Author Response

We thank the reviewer for outlining this mistake; we have corrected it as indicated. 
